# Performance Characteristics and Limitations of the Available Assays for the Detection and Quantitation of Monoclonal Free Light Chains and New Emerging Methodologies

**DOI:** 10.3390/antib13010019

**Published:** 2024-03-11

**Authors:** Hannah V. Giles, Kamaraj Karunanithi

**Affiliations:** 1Department of Clinical Haematology, University Hospitals Birmingham NHS Foundation Trust, Birmingham B15 2SY, UK; 2Instute of Immunology and Immunotherapy, University of Birmingham, Birmingham B15 2TT, UK; 3Department of Clinical Haematology, University Hospitals North Midlands NHS Trust, Royal Stoke Hospital, Newcastle Road, Stoke-on-Trent ST4 6QG, UK; kamaraj.karunanithi@uhnm.nhs.uk; 4School of Medicine, Keele University, Keele, Newcastle-under-Lyme ST5 5BG, UK

**Keywords:** immunofixation electrophoresis, urine protein electrophoresis, serum FLC assays, mass spectrometry

## Abstract

Light chain measurements form an essential component of the testing strategy for the detection and monitoring of patients with suspected and/or proven plasma cell disorders. Urine-based electrophoretic assays remain at the centre of the international guidelines for response assessment but the supplementary role of serum-free light chain (FLC) assays in response assessment and the detection of disease progression due to their increased sensitivity has been increasingly recognised since their introduction in 2001. Serum FLC assays have also been shown to be prognostic across the spectrum of plasma cell disorders and are now incorporated into risk stratification scores for patients with monoclonal gammopathy of undetermined significance (MGUS), smouldering multiple myeloma, and light chain amyloidosis (AL amyloidosis), as well as being incorporated into the criteria for defining symptomatic multiple myeloma. There are now multiple different commercially available serum FLC assays available with differing performance characteristics, which are discussed in this review, along with the implications of these for patient monitoring. Finally, newer methodologies for the identification and characterisation of monoclonal FLC, including modifications to electrophoretic techniques, mass spectrometry-based assays and Amylite, are also described along with the relevant published data available regarding the performance of each assay.

## 1. Introduction

Plasma cell dyscrasias encompass a broad range of disorders, from the pre-malignant condition MGUS to AL amyloidosis, which is typically associated with a low tumour burden, and malignant multiple myeloma in which the tumour burden is much higher [1,2]. Plasma cells secrete antibodies; and in most patients with a plasma cell dyscrasia, a monoclonal immunoglobulin is detectable in the serum and/or urine due to secretion of a single type of antibody by the clonal plasma cell population. A typical antibody is composed of two identical heavy chains and two identical light chains; and even in health, the light chains are secreted in slight excess in comparison to heavy chains, which results in low levels of circulating polyclonal light chains. However, in patients with plasma cell disorders, the FLC are frequently produced in significant excess and the resultant circulating monoclonal light chains can play an important role in the clinical sequelae of some of the plasma cell disorders, such as AL amyloidosis. Due to somatic hypermutation and VDJ rearrangement, the monoclonal immunoglobulin produced by each patient’s plasma cell dyscrasia has a unique structure and biochemical properties, which enable it to be used as a biomarker to assess for signs of progression in collaboration with assessments for end-organ damage and also to track the response to clonally directed treatment [3].

In 80–85% of patients with multiple myeloma and MGUS, the monoclonal protein produced by the clonal plasma cell population produces an intact immunoglobulin monoclonal protein, most commonly IgG, and 15–20% of patients have a FLC-only monoclonal protein, which may be of either free kappa or free lambda light chain isotype, without a corresponding heavy chain [4,5,6,7]. Light chain-only monoclonal proteins are slightly more common in the rare condition AL amyloidosis, where they are found in 25–30% of patients [8]. Even in patients with an intact immunoglobulin monoclonal protein, excess FLC production is frequently observed [5], and may become the predominant or only monoclonal protein detectable at disease progression [9,10,11].

In AL amyloidosis, it is the aggregation and deposition of the structurally abnormal light chains that leads to progressive organ involvement. Therefore, assays that can detect and monitor changes in the levels of monoclonal light chains are key in the detection and monitoring of patients with this rare plasma cell disorder. Excess production of monoclonal FLC can also lead to direct organ toxicity in patients with multiple myeloma as high levels of circulating monoclonal FLC can cause renal impairment due to cast nephropathy. The risk of myeloma-induced renal impairment increases as the level of circulating monoclonal FLC increases; and therefore, tests that can rapidly identify and quantitate monoclonal light chains are crucial for the early identification of acute renal impairment due to multiple myeloma, which is most commonly due to cast nephropathy, and is rarely seen with FLC levels < 500 mg/L [12,13]. The identification and prompt initiation of systemic anti-myeloma treatment in patients with severe renal impairment due to cast nephropathy are essential to maximise the chances of renal function recovery, which can be achieved in over half of the affected patients if a rapid reduction in the monoclonal light chain level is achieved [14]. 

Over the last two decades, there have been several developments in the tests available for the detection and monitoring of monoclonal FLC in patients with plasma cell dyscrasias, including refined definitions of measurable disease using these assays across the spectrum of plasma cell dyscrasias and response criteria incorporating them. As our understanding of the strengths and limitations of the commonly used assays has evolved, some areas of controversy, such as how to interpret serum FLC ratio results in patients with renal impairment, have emerged. There have also been extensive efforts made to identify novel assays and modifications of the existing assays that can enhance sensitivity given the deep responses obtained in the majority of patients with plasma cell disorders treated with modern chemotherapy and immunotherapy regimens [15,16,17,18,19,20]. In this review, we summarise these developments as well as evaluating the areas of controversy. In addition, newer methodologies that have emerged recently are described, including mass spectrometry-based assays, modifications to the existing electrophoretic techniques and Amylite, and we review the strengths and limitations of these assays in comparison to standard assays used for the detection and monitoring of monoclonal FLC in patients with plasma cell dyscrasias.

## 2. Laboratory Assays for the Detection and Measurement of Monoclonal FLC in Patients with Plasma Cell Dyscrasias and Their Strengths and Limitations

### 2.1. Urine Protein Electrophoresis and Urine Immunofixation for the Detection and Measurement of Monoclonal FLC (Bence Jones Protein) in Urine 

Bence Jones Proteins (BJP) are monoclonal immunoglobulin light chains, found in the urine of patients with plasma cell dyscrasias that were initially described by Henry Bence Jones in 1845 [21]. They are the first cancer biomarker to have been discovered. BJP was confirmed to be monoclonal in nature following the observation by Korngold and Lipari in 1956 that they were found in the urine of several patients with multiple myeloma and it was noted belonged to two classes: kappa light chain BJP and lambda light chain BJP [22]. The evolution of the tests available for the detection and monitoring of monoclonal light chains in patients with plasma cell disorders is summarised in Figure 1.

Electrophoresis was first applied to the study of multiple myeloma in 1939 [23] and its sensitivity was further enhanced by the development of immunofixation electrophoresis (IFE) and direct immuno-electrophoresis by Grabrar and Williams in 1953 [24]. Drs Edelman and Porter were awarded the Nobel Prize in Physiology or Medicine in 1962 for their work on the structure of antibodies in which they demonstrated that the light chains of the monoclonal protein identified in the serum of a patient with multiple myeloma were identical to the BJP detected in their urine. Five years later, Dr Putnam demonstrated that BJPs have distinct peptide sequences which lead to specific mobility patterns [25]. Since these discoveries, urine protein electrophoresis and IFE of urine samples utilising antisera specific for total kappa and total lambda light chains in combination with heavy chain specific antisera have been providing a relatively simple and low-cost methodology for the detection and quantitation of BJP with a limit of sensitivity of 10–40 mg/L [27,28,29]. 

The American College of Pathologists and the International Myeloma Working Groups recommend that 24 h urine samples are used for BJP testing [30,31]. However, 24 h urine collections are cumbersome and impractical for many elderly frail patients, so the IFCC Committee on Plasma Proteins and the SIBioC Study Group on Protein recommends using second void of the day urine samples as spot samples and that the concentration of BJP is expressed relative to urinary creatinine [28]. However, all of the current internationally recognised response criteria for patients undergoing treatment for plasma cell dyscrasias incorporate BJP quantitation in mg/24 h [32,33,34], so there is often differing practice in how patients are monitored in real-world practice compared to those treated in clinical trials. 

Singh et al. have recently demonstrated that using FLC-specific antisera for the detection of BJP in urine samples enhances the sensitivity of IFE for residual disease detection [35]. However, this technique is not in routine clinical use and the standard total light chain assays remain the recommended methodology.

### 2.2. Modifications to Enhance the Sensitivity of Serum IFE

Standard electrophoretic assays lack sensitivity for the detection of lower-level monoclonal FLC as they only include total light chain reagents. Small FLC monoclonal peaks may therefore frequently be masked by the total light chain background from the intact immunoglobulin polyclonal background which is measured in g/L in contrast to FLC which are present in mg/L [36,37]. Although electrophoretic analysis of urine rather than serum overcomes the issue of the polyclonal intact immunoglobulin background, the sensitivity of these assays is limited by the fact that the monoclonal FLC needs to be secreted in sufficient excess for it to overwhelm the renal re-absorptive capacity. Techniques that could enhance the sensitivity of serum IFE, including using size-exclusion ultrafiltration and employing FLC-specific antisera, have therefore been explored and encouraging results with sensitivity down to 1 mg/L have been reported [38,39]. However, at present, the suitability for large-scale use of these assays has not been explored and they only represent assays in the research and development phase.

### 2.3. Serum FLC Assays

#### The Spectrum of Commercially Available Serum FLC Assays

The introduction of serum assays that could specifically measure FLC at the beginning of the twenty-first century represented a significant advance in the diagnostic armamentarium for patients with plasma cell dyscrasias. These assays utilise antibodies that specifically target epitopes on light chains which are only accessible in circulating light chains that are not bound to heavy chain hence they specifically measure only FLC (Figure 2). These assays measure the total quantity of serum-free kappa and serum-free lambda light chains and rely on the indirect measure of a skewed ratio between the two FLC isotypes to infer the presence of clonal FLC production [26,40]. 

The antisera used in FLC assays bind to a portion of light chain constant domain, which is hidden when light chains are bound to heavy chains as part of intact immunoglobulins. This means that these reagents specifically bind FLC in contrast to total light chain reagents, which are used in standard serum IFE assays, that bind both FLC and light chains bound to heavy chains.

The greater sensitivity of these assays was first demonstrated by Drayson et al., who demonstrated that an increased involved FLC and an abnormal FLC ratio were present in 19/28 patients from UK myeloma trials who had previously been categorised as having non-secretory multiple myeloma [26]. Their utility in monitoring patients with oligo-secretory myeloma and FLC multiple myeloma led to these assays being incorporated into the 2006 IMWG Uniform Response Criteria [41] and the criteria for response assessment for patients undergoing treatment for systemic AL amyloidosis [42]. Serum FLC assays have also subsequently been incorporated into the diagnostic criteria for FLC MGUS [4] and risk stratification models for MGUS, smouldering multiple myeloma, and AL amyloidosis [43,44,45,46]. 

Since the first commercially available serum FLC assay, Freelite, was launched by the Binding Site Ltd. (Birmingham, United Kingdom) in 2001, multiple other commercially available serum FLC assays have been released (Table 1). Similar to the Freelite assay, the KLoneus assay, the Diazyme FLC assay and the Sebia FLC assay all use polyclonal antisera [47,48,49]. In contrast, the N-Latex FLC assay developed by Siemens and the Seralite assay developed by Abingdon Health both use monoclonal antisera [48]. Polyclonal and monoclonal antisera have been shown to have differential affinities for monomers and dimers, which can lead to discrepant results being obtained between the assays. In addition, these assays have been shown to have significant lot-to-lot variation, non-linearity in the presence of antigen excess, variable performance when they are used on different testing platforms and variable FLC polymerisation can lead to overestimation of serum FLC concentrations [50,51,52]. As no international reference material for the measurement of FLC is available, it is not possible to ascertain which method is the most accurate [53]. Due to fact that the results obtained by each of the assays are not directly comparable [36,50,54,55,56,57,58,59,60], it is essential that patients are monitored throughout their treatment journey using the same methodology. 

These inter-assay quantitative discrepancies therefore affect the application of risk scores that include absolute FLC values derived from studies using the Freelite assay, such as the Mayo Amyloidosis risk score [43] and response criteria with specific FLC value thresholds such as an amyloid very good partial response, which is defined as a difference between the uninvolved and involved FLC of <40 mg/L [33].

The inter-assay discrepancies have also been shown to affect the kappa lambda ratio results. Therefore, the FLC ratio thresholds that have been incorporated into the risk models and response criteria, which are all based on studies utilising the Freelite assay, may not be applicable when the other serum FLC assays are employed [56]. This is most likely to relevant for the risk models for MGUS [63] and smouldering multiple myeloma [46] and the criteria for a stringent complete response where the FLC ratio thresholds are lower. However, they may also make subtle differences in some patients where the only criterion for symptomatic multiple myeloma fulfilled is the FLC ratio being ≥100 [50]. 

There are also several areas of controversy in relation to the application of serum FLC results for disease classification and response assessment. Firstly, the FLC ratio criterion as a biomarker of malignancy which defines symptomatic multiple myeloma has been the subject of extensive discussion since its introduction in 2014, owing to the fact that several studies have found a much lower risk of progression to symptomatic disease than found in the earlier studies which were used as the basis for this criterion being included in the SLiM CRAB criteria [31,64,65,66,67,68]. The Mayo group have shown that BJP assessment alongside FLC ratio results may further help refine the risk stratification but even in patients with a serum FLC ratio ≥ 100 and BJP ≥ 200 mg/24 h, the two-year risk of progression was significantly lower than 80% at 36.2% [65]. The lower risk of progression found in the more recent studies may be at least in part reflective of the greater sensitivity of modern cross-sectional imaging techniques for the detection of earlier myelomatous bone lesions [69]. Another limitation of this criterion is that some patients have significant changes in their FLC ratio results between samples due to changes in the level of the uninvolved FLC in the absence of any significant change in the level of the involved FLC. It is therefore important to look at the whole result panel and carefully examine the trends in all parameters in addition to the clinical status of the patient before making decisions about possible disease progression.

Secondly, renal impairment makes the interpretation of serum FLC assay results more complicated as FLC clearance is slower in the presence of renal impairment, which leads to higher polyclonal light chain levels. The relative clearance rates of kappa and lambda light chains also differs between the kidneys and the reticuloendothelial system. Therefore, in patients with renal impairment the normal ratio of kappa to lambda differs from that seen in patients with normal renal function [40]. Several groups have proposed modified renal reference ranges to avoid false positive results [40,70,71]; however, to date, none of these modified reference ranges have been externally validated and or formally incorporated into response criteria. The use of these modified renal reference ranges not only reduces the rate of false positives for low level FLC abnormalities but also improves the sensitivity of serum FLC assays for the detection of low-level lambda FLC monoclonal proteins [40,72,73]. Due to the lack of international consensus on which reference range should be used, there is variable practice with regard to the interpretation of FLC value and ratio results between institutions.

Thirdly, treatment-related immune suppression and oligoclonal immune reconstitution have been shown to cause false positive results, particularly low-level abnormal ratios skewed toward kappa [5,74,75]. These false positive results may at least partially explain the varying results with regard to the prognostic significance of FLC ratio normalisation in IFE-negative patients [75,76,77,78]. Many groups, including most recently the iStopMM group, have proposed modifications to the serum FLC ratio reference range published by Katzmann et al. [61,79,80,81]. These amended reference ranges may help to reduce false positive rates and refine response criteria and the diagnostic criteria for FLC MGUS, but they need to be externally validated. There is a clear need for updated guidance on which reference ranges should applied in different patient cohorts and a review of how this may impact the currently utilised response criteria.

Although many centres have transitioned over to using serum FLC testing in place of BJP assessments in routine practice, it is important to acknowledge that the monoclonal protein size criteria for the differentiation between MGUS and multiple myeloma is based on the BJP concentration being ≥ 500 mg/24 h [31] and that there is no recommended serum FLC level that by definition differentiates between MGUS and myeloma. In centres where BJP analysis is not regularly performed, it is crucial to establish clear guidelines for assessing the bone marrow in patients with abnormal FLC results. This is important, even if there is no organ damage, to distinguish between MGUS and myeloma based on the percentage of plasma cell infiltration in the bone marrow. It is crucial to differentiate between these conditions to ensure that the follow-up intervals are appropriate given the differential risk of progression to symptomatic multiple myeloma between these two conditions (0.5–1% per year for MGUS versus 10% per year for smouldering multiple myeloma) [44].

### 2.4. Mass Spectrometry-Based Assays for the Detection of Monoclonal FLC

Mass spectrometry assays have emerged as a new methodology for the detection and monitoring of patients with plasma cell dyscrasias over the last decade. Two types of assays have been developed a clonotypic peptide approach and an intact light chain approach (Figure 3). The clonotypic peptide approach utilises unique peptide sequences from the complementarity determining region of the monoclonal protein to provide biomarkers which can be used to monitor treatment response. The clonotypic peptide approach is highly personalised and sensitive methodology, with a limit of detection of 0.5–1.0 mg/L for both intact immunoglobulin [82,83] and FLC-only monoclonal proteins [84]. However, it is a labour-intensive technique with much lower throughput than the currently available serum FLC and electrophoretic assays. In addition, there are a small proportion of patients in whom suitable peptide sequences cannot be identified [82] and it has been noted that the pool of unique targets is lower when only light chains are available for sequencing [85]. Owing to the lower throughput of these assays, the currently published studies exploring the potential utility of the clonotypic peptide approach have been small in nature and have not yet demonstrated a statistically significant association between patients survival and depth of response assessed using clonotypic assays but they consistently show much higher sensitivity than the electrophoretic assays [82,83,84,85,86].

The intact light chains assays utilise the principal that the light chain of the monoclonal protein will have a specific isotype and unique mass due to its unique amino acid sequence which helps to track the monoclonal protein during the course of treatment. Samples undergo an immune enrichment step to remove non-immunoglobulin proteins, then captured immunoglobulins are eluted and reduced, so that the light chain portion of the monoclonal protein can be analysed by mass spectrometry (Figure 3). The eluates derived from the immune enrichment step can be analysed by either liquid chromatography-mass spectrometry (LC-MS) or matrix-assisted laser desorption/ionisation time-of-flight mass spectrometry (MALDI-TOF MS) [3]. Analysis by LC-MS provides greater sensitivity [87,88,89,90,91] but is much more time-consuming as manual steps are required for the transfer of the eluates into suitable samples vials for analysis and the data analysis. Sample throughput is therefore much lower for the LC-MS-based intact light chain assays. The upfront equipment costs are also much higher, so the suitability of these assays for high-throughput clinical analysis is uncertain and, at present, the LC-MS-based intact light chain assays represent a research tool only.

**Figure 3 antibodies-13-00019-f003:**
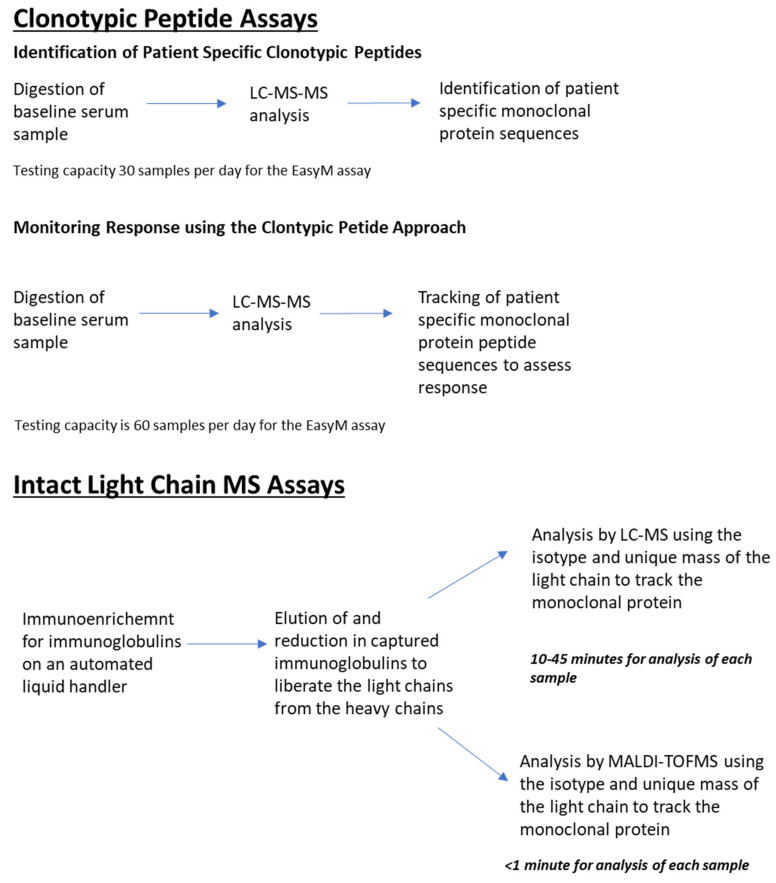
Workflow for the clonotypic and intact light chain MS-based assays [82,92].

Owing to their higher-throughput capabilities the MALDI-TOF MS intact light chains assays have been much more widely explored in clinical studies and following several studies encouraging results their use as an alternative to immunofixation electrophoresis was approved by the International Myeloma Working Group in 2021 [3]. There are currently two MALDI-TOF MS-based assays available for clinical use. The first is the MASS-FIX assay, which was developed by the Mayo clinic, and has replaced serum immunofixation electrophoresis at their institution [93]. The Mayo clinic is the only institution where this assay is available. The MASS-FIX assay is used as a qualitative assay, which is interpreted in a binary fashion, and employs antisera specific for IgG, IgA, IgM, total kappa and total lambda and does not contain any FLC-specific reagents. It has been shown to provide greater sensitivity than immunofixation electrophoresis and has been shown to be prognostic in patients undergoing treatment for newly diagnosed multiple myeloma [94] and also provides additive prognostically relevant information in patients who have no detectable minimal residual disease by flow cytometry [95]. 

Although the MASS-FIX assay has been shown to be able to detect low-level residual disease in some patients with AL amyloidosis [91], other studies have shown that there are some patients in whom low-level monoclonal FLC production is detectable using serum FLC and/or electrophoretic analysis of the urine [39,96]. The differential sensitivity likely reflects the fact that patients the sensitivity of the mass spectrometry assay will depend on the location of the monoclonal protein in relations to the bell curve distribution of the polyclonal background. These results highlight that even when these sensitive intact light chain assays are employed, the use of FLC-specific tests is still required to ensure low-level FLC-only disease is not missed if mass spectrometry testing only utilises serum samples and total light targeting reagents. Recently, Moonen et al. have demonstrated that MASS-FIX assay can be applied to the analysis of urine samples and that its sensitivity using unconcentrated urine samples is similar to that of urine immunofixation analysis of 100× concentrated urine samples [97]. To date, urine MASS-FIX analysis has not been included in any of the large published clinical studies exploring the prognostic significance of MASS-FIX status.

EXENT (formerly known as QIP-MS) is the first commercialised MALDI-TOF MS-based intact light chain assay that has gained approval for clinical use. At present, this assay is approved for use within Europe, but FDA approval is awaited. Similar to the MASS-FIX assay, the standard EXENT assay does not include FLC-specific reagents [98]. However, there are some methodological differences between the assays. Firstly, the EXENT assay utilises antisera conjugated to magnetic microparticles, whereas the MASS-FIX assay uses camelid nanoparticles [93,98]. There are no published head-to-head comparisons of the sensitivity of the two assays, so it is unclear what impact the difference in reagents has on assay performance. Secondly, the software for the EXENT assay automatically interprets the mass spectra and provides quantitative results for intact immunoglobulin monoclonal proteins in contrast to the qualitative results provided by the MASS-FIX assay. Quantitation is performed by analysing using proprietary software which automatically performed the quantitative analysis using a method similar to that previously described by Mills et al. [99]. The reported sensitivity for this assay for intact immunoglobulins monoclonal proteins is 15 mg/L [98]; however, as there is no assay that specifically quantitates total kappa and total lambda, the EXENT assay can only detect the presence of FLC-only monoclonal proteins but cannot quantitate them. At present, the widespread use of the intact light chain MALDI-TOF MS-based assays is limited as only a few centres have the relevant instrumentation and the assay cost is much higher than the electrophoretic assays [92]. A summary of the advantages and disadvantages of the MS-based assays in comparison to the assays in wider routine clinical use is listed in Table 2.

Light chain N-linked glycosylation of the variable region of monoclonal light chains is a post-translational modification which is easily identified using the intact light chain assays based on the presence of polytypic peaks with an increased mass-to-charge ratio compared to the typical mass-to-charge range for kappa and lambda light chains [104] (Figure 4). Light chain N-linked glycosylation has been shown to be found more commonly in patients with AL amyloidosis in comparison to patients with multiple myeloma and MGUS [105,106] and has therefore emerged as an area of interest which may assist in the earlier identification of patients with AL amyloidosis. In contrast to heavy chains, in which glycosylation is a normal feature, light chain glycosylation has not been observed in serum samples from healthy donors which have been used as control samples in the development of these assays. The presence of N-linked glycosylation was also reported to be associated with an increased risk of progression from MGUS to a symptomatic plasma cell disorder in a study by Dispenzieri et al. [107]; however, there are only limited data on its prognostic significance in patients undergoing treatment for symptomatic plasma cell disorders [108].

A companion assay, referred to as FLC-MS, which employs polyclonal antisera specific for free kappa and free lambda light chains, has been developed and explored in a number of clinical studies [101,109,110,111,112,113,114,115,116]. It has been shown to provide greater sensitivity for the detection of disease both at presentation and following treatment in patients with systemic AL amyloidosis [115,116]. In the study by Bomsztyk et al., the presence of residual monoclonal FLC detectable by FLC-MS was associated with reduced overall survival, even when the analysis was restricted to patients in an amyloid complete haematological response [116], emphasising the high sensitivity of this technique. This assay has the advantage over the serum FLC ratios that it specifically tracks the monoclonal FLC through its *m*/*z* and therefore enables differentiation between oligoclonal peaks and low-level residual disease, reducing the likelihood of false positive results due to oligoclonal immune reconstitution.

The FLC-MS assay has also been shown to provide additional sensitivity when used in collaboration with the five-bead EXENT assay in patients undergoing treatment for newly diagnosed multiple myeloma [101,106,111]. Although the results of these early clinical studies exploring the potential clinical utility of FLC-MS look encouraging, this assay is still in a research and development phase and is not available for clinical use and does not yet have software available for the automated stacking and interpretation of the mass spectra.

### 2.5. Isoelectric Focussing

Isoelectric focussing is already used in many labs for the detection of oligoclonal bands in the cerebrospinal fluid using commercially available kits and is a relatively inexpensive technique. The potential utility of isoelectric focusing followed by affinity immunoblotting was initially explored in plasma cell disorders in the 1980s but it was never widely adopted despite promising results [117,118,119]. Recently, Zeman et al. have published a study exploring the utility of this technique in patients with multiple myeloma and AL amyloidosis and have shown it can provide high sensitivity for the detection residual monoclonal FLC in the serum (sensitivity 0.05–0.1 mg/L for kappa FLC and 0.1–0.2 mg/L for lambda FLC). Like the electrophoretic and MS-based assays, this assay specifically tracks the monoclonal FLC and this enables it to differentiate between patients with an abnormal serum FLC ratio due to persistent disease and patients with abnormal serum FLC ratios, which are likely due to oligoclonal immune reconstitution following intensive plasma cell-directed therapies such as autologous stem cell transplantation. However, it is important to note that ambiguous results were observed in 4–11% of samples [73].

### 2.6. Amylite

Amylite is a novel diagnostic test that has been developed to facilitate the identification and quantification of amyloidogenic lambda light chains. Jiang et al. reported that following limited proteolysis using Proteinase K, samples from patients with AL amyloidosis yielded a 23 kDa fragment that was composed of the homodimeric light chain domain (dLCCD) linked by a disulfide bond. Using monoclonal antibodies specific for a cryptic epitope at the N-terminus of the fragment the dLCCD can be detected and quantitated by Western blotting or Meso Scale Discovery analyses. In this study, dLCCD were detected in 67/70 samples from patients with AL amyloidosis and none of the samples from healthy donors. This assay therefore has great potential to assist in the early detection of lambda light AL amyloidosis, which represents 75% of cases of AL amyloidosis [120]. This assay may be valuable in assisting in the earlier detection of AL amyloidosis; however, these early results need to be validated in larger studies.

## 3. Conclusions and Future Directions

FLC measurements form a key component of the monitoring of patients all patients with plasma cell disorders, not only those with FLC-only disease. The introduction of serum-based FLC assays has provided enhanced sensitivity [121] and reduced the number of patients with non-measurable disease. However, this is still an issue for 1–4% with multiple myeloma [26,36,122,123] and approximately 20% of patients with AL amyloidosis [124]. These assays provide valuable prognostic information in newly diagnosed patients and are highly valuable for assisting in the rapid identification of patients with renal impairment due to cast-nephropathy [12,102]. However, due to the fact that they do not directly measure the monoclonal FLC there are issues with false positive results [74,78] and recent studies have questioned the prognostic significance of FLC ratio normalisation in IFE-negative myeloma patients [78,125,126]. Several modified reference ranges for the Freelite assay, which is the most commonly used serum FLC assay, have been proposed over the years but none have been incorporated into the internationally recognised response criteria. Further work is urgently needed to evaluate the optimal reference range to use, taking into account the effects of age, ethnicity and renal function, and then the thresholds within the current risk models and response criteria should be reviewed. This work will be essential to ensure the optimal balance between sensitivity and limiting the number of false positives can be achieved.

The intact light chain MS-based assays, which provide a high-throughput methodology, for the sensitive measurement of monoclonal light chains may provide a better methodology in the future. They also have the potential to improve the proportion of patients with serologically measurable disease given their higher sensitivity [109]. However, their widespread application is currently limited as only a few centres have the instrumentation required to perform the testing and they are more expensive. For the foreseeable future it may therefore be more practical to only reflex to these more sensitive tests once negativity using the cheaper and more widely available electrophoretic assays and immunoassays has been achieved. It is also important to recognise that although the MS assay provide greater sensitivity for the detection of monoclonal proteins overall, FLC-specific testing remains important when only the five bead assays containing total light chain but no FLC-specific reagents are used. This is due to the fact that patients with low-level monoclonal FLC production may be negative by MS but have disease detectable using serum FLC assays and/or by electrophoretic analysis of the urine [96].

## Figures and Tables

**Figure 1 antibodies-13-00019-f001:**
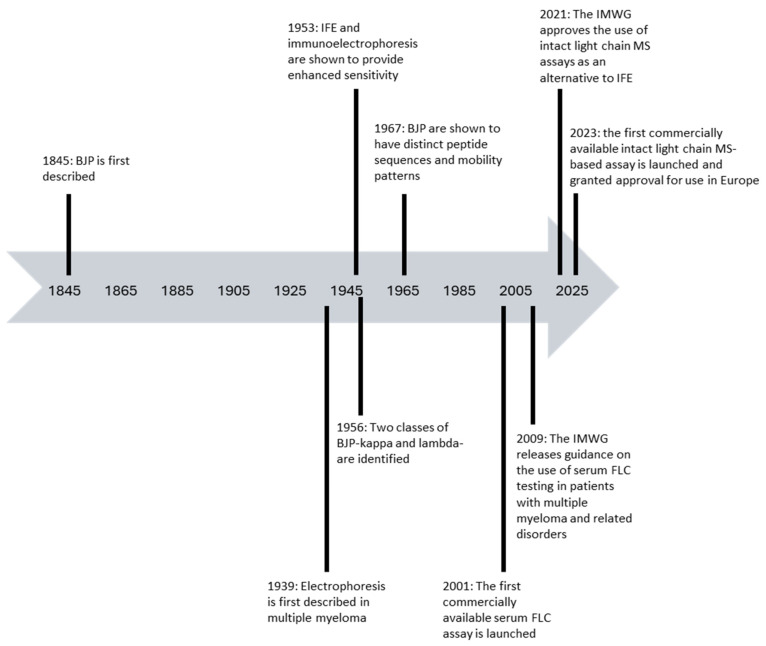
Timeline of the evolution of tests available for the detection of monoclonal light chains in patients with plasma cell disorders [3,5,21,22,23,24,25,26].

**Figure 2 antibodies-13-00019-f002:**
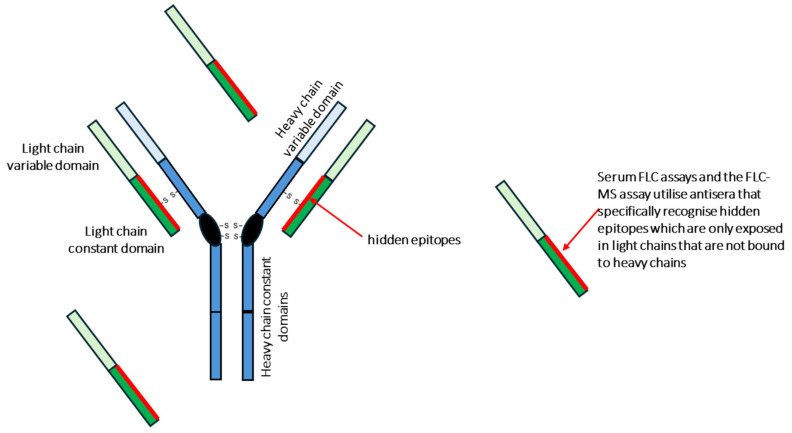
The principle of FLC-specific assays.

**Figure 4 antibodies-13-00019-f004:**
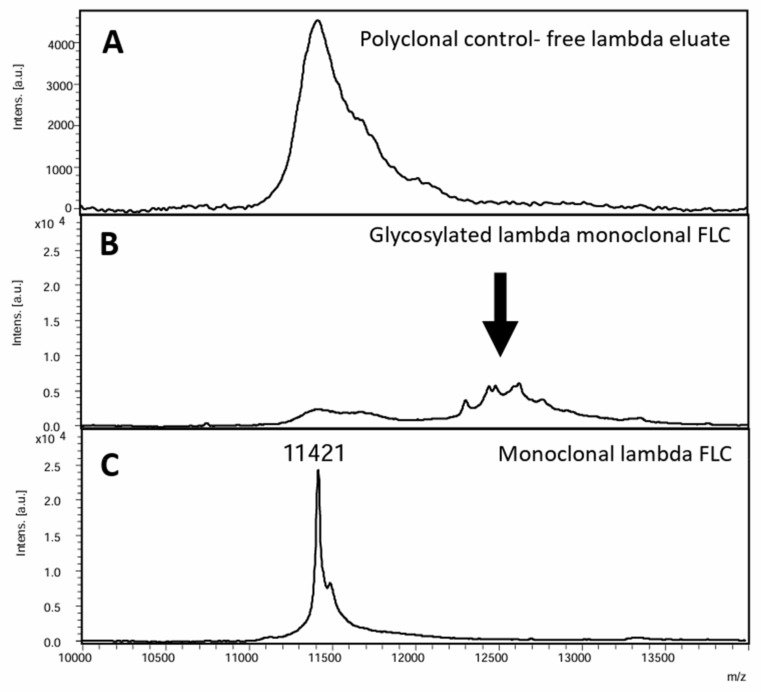
Intact light chain MS-based assays can detect light chain glycosylation. (**A**) shows an example mass spectrum from a polyclonal sample that has undergone immune enrichment for lambda FLC and then analysis by MALDI-TOF MS. (**B**) shows an example mass spectrum from a patient with a glycosylated monoclonal lambda FLC showing a polytypic hedgehog shaped peak with an increased *m*/*z* compared to the standard *m*/*z* region for the double charged light chains (highlighted by the black arrow). (**C**) shows an example mass spectrum from a sample with a lambda FLC monoclonal peak at *m*/*z* 11,421 for the doubly charged light chain peak. This figure is derived from a sample from the ALchemy study [103].

**Table 1 antibodies-13-00019-t001:** Summary of the Commercially Available Serum FLC Assays.

Assay Name	Antisera	Testing Methodology	FLC Reference Ranges (mg/L)	FLC Ratio Reference Range
Freelite [61,62]	Polyclonal	Turbidimetry/ nephelometry	Kappa 3.3–19.4	0.26–1.65
			Lambda 5.7–26.3	
Sebia FLC [55]	Polyclonal	ELISA	Kappa 6.4–17.4	0.46–1.51
			Lambda 8.4–21.8	
Diazyme [47,62]	Polyclonal	Turbidimetry	Kappa 2.37–20.73	0.22–1.74
			Lambda 4.23–27.69	
Kloneus Free Light Chain [62]	Polyclonal	Turbidimetry/nephelometry	Kappa 3.3–19.4	0.26–1.65
			Lambda 5.7–26.3	
Seralite [57]	Monoclonal	Competitive inhibition	Kappa 5.3–22.7	0.5–2.5
		immunochromatography	Lambda 4.0–25.1	
N Latex FLC [57]	Monoclonal	Nephelometry	Kappa 6.7–22.4	0.31–1.56
			Lambda 8.3–27.0	

**Table 2 antibodies-13-00019-t002:** Summary of the advantages and disadvantages of the currently available assays for the detection and monitoring of monoclonal light chains.

Method	Approved Applications	Advantages	Disadvantages
SPE and sIFE	Monitoring patients with MGUS, multiple myeloma, and AL amyloidosis [31,33]	Widely available	Low sensitivity compared to FLC-specific assays [26]
		Relatively low cost	Not used for quantitative assessment of FLC monoclonal proteins
uPE and uIFE	Response assessment in patients with multiple myeloma and AL amyloidosis [31,33]	Widely available	False positives for BJP in patients with chronic kidney disease [100]
	≥500 mg/24 h BJP differentiates myeloma from MGUS in the IMWG diagnostic criteria [31]	Relatively low cost	Requires a 24 h urine collection
Serum FLC assays	Response assessment in patients with multiple myeloma and AL amyloidosis [5,31,33]	Widely available	No international consensus on the most appropriate reference range to use in patients with varying degrees of renal impairment
	Risk stratification of patients with MGUS, SMM and AL amyloidosis [43,44,45]	Provides additional sensitivity for the detection of low-level monoclonal FLC in patients with FLC myeloma and AL amyloidosis [5,26,36]	Rely on the ratio between the uninvolved and involved FLC as an indirect indicator of clonality, which can lead to false positives in the presence of oligoclonal immune reconstitution and/or treatment related immune suppression [101]
	The level of the involved FLC helps identify patients with myeloma and renal impairment who are likely to have renal impairment due to cast nephropathy [12,13,102]	Automated sample processing and result generation	
	FLC ratio ≥100 is incorporated into the SLiM CRAB criteria for the identification of patients with symptomatic multiple myeloma [30]	Convenient -analysis can be performed on the same sample used for SPE and sIFE	
Intact light chain MS assays	In lieu of immunofixation in patients with multiple myeloma and related disorders [3]	Greater sensitivity compared to sIFE [91,94,103]	More expensive than the electrophoretic assays and serum FLC assays [92]
		Tracking the monoclonal protein using its isotype and mass-to-charge ratio enables more reliable differentiation between oligoclonal peaks and residual low-level monoclonal protein monoclonal protein	The availability of the intact light chain assays is limited: EXENT is currently only approved for use in Europe; MASS-FIX is only available for use in Europe; and FLC-MS is not currently approved for clinical use
		High sample throughput possible due to automated sample processing and semi- or fully automated result interpretation depending on the assay used [93]	Risk of missing low-level FLC-only monoclonal protein if sFLC and uIFE are not used alongside MS assays that only include total light chain-specific reagents [96]
		Able to identify post-translational modifications such as N-linked glycosylation [92,96,103]	MASS-FIX and EXENT only provide a qualitative assessment about the presence absence of monoclonal light chains (FLC-MS could provide quantitative assessments but is currently only a research tool)

SPE: serum protein electrophoresis, sIFE: serum immunofixation electrophoresis, uPE: urine protein electrophoresis, uIFE: urine immunofixation electrophoresis, SMM: smouldering multiple myeloma, and MS: mass spectrometry.

## Data Availability

Not applicable.

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
