# Peer review of "Performance Characteristics and Limitations of the Available Assays for the Detection and Quantitation of Monoclonal Free Light Chains and New Emerging Methodologies"

_2073-4468, 2024, doi:10.3390/antib13010019_

Round 1
Reviewer 1 Report
Comments and Suggestions for Authors
This is a pretty extensive review on a topic that is not that well reviewed. The authors review methods and their performance characteristics for serum free light chain detection for monoclonal antibodies. Overall, the paper is worthy of publication, with the following suggested revisions:
1. The authors describe all the methods very qualitatively - some aspects on quantitation and limitations based on that can be described. Right now, it comes off a drab-and-dull reading essay.
2. A table to summarize the limitations of all the methods may help.
Author Response
Dear Reviewer One,
Thank you very much for taking the time to review this manuscript and providing constructive comments. We have addressed your suggestions and feel that these have helped us to improve the quality of our review article.
Reviewer one comments and responses:
This is a pretty extensive review on a topic that is not that well reviewed. The authors review methods and their performance characteristics for serum free light chain detection for monoclonal antibodies. Overall, the paper is worthy of publication, with the following suggested revisions:
- The authors describe all the methods very qualitatively - some aspects on quantitation and limitations based on that can be described. Right now, it comes off a drab-and-dull reading essay.
-we have included additional details of how quantitative elements of the assays are applied in the classification, staging and monitoring of patients with multiple myeloma and related plasma cell disorders as follows:
“The risk of myeloma-induced renal impairment increases as the level of circulating monoclonal FLC increases and therefore tests that can rapidly identify and quantitate monoclonal light chains are crucial for the early identification of acute renal impairment due to multiple myeloma, which is most commonly due to cast nephropathy, and is rarely seen with FLC levels <500mg/L (12, 13). The identification and prompt initiation of systemic anti-myeloma treatment in patients with severe renal impairment due to cast nephropathy is essential to maximise the chances of recovery renal function, which can be achieved in over half of affected patients if rapid reduction of the monoclonal light chain level is achieved (14)”
“These inter-assay quantitative discrepancies therefore affect the application of risk scores that include absolute FLC values derived from studies using the Freelite assay, such as the Mayo Amyloidosis risk score (41) and response criteria with specific FLC value thresholds such as an amyloid very good partial response, which is defined as a difference between the uninvolved and involved FLC of <40mg/L (30).”
“Although many centres have transitioned over to using serum FLC testing in place of BJP assessments in routine practice, it is important to acknowledge that the monoclonal protein size criteria for the differentiation between MGUS and multiple myeloma is based on BJP concentration being ≥500mg/24 hours (28) and that there is no recommended serum FLC level that by definition differentiates between MGUS and myeloma. It is therefore important that in centres that are not routinely performing BJP analysis that clear criteria for bone marrow assessment in patients with abnormal FLC results is defined even in the absence of end organ damage to allow differentiation between MGUS and myeloma based on the percentage of plasma cell infiltration in the bone marrow. It is important to differentiate between these conditions to ensure that the follow-up intervals are appropriate given the differential risk of progression to symptomatic multiple myeloma between these two conditions.”
“The reported sensitivity for this assay for intact immunoglobulins monoclonal proteins is 15 mg/L (94), however as there is no assay that specifically quantitates total kappa and total lambda, the EXENT assay can only detect the presence of FLC only monoclonal proteins but cannot quantitate them.”
We have included two additional figures to break up the text further and also offered further details on the clinical relevance of some of the areas of controversy in the application of the different assays to enhance interest for readers.
- A table to summarize the limitations of all the methods may help.
We have added a table summarizing the applications, advantages and disadvantages of the approved methods for the detection and monitoring of monoclonal light chains in patients with multiple myeloma and related plasma cell disorders. This is table 2 in the revised manuscript.
Thank you for taking the time to review this revised manuscript.
Yours sincerely
Dr Hannah Giles and Professor Karunanithi
Reviewer 2 Report
Comments and Suggestions for Authors
In the manuscript entitled ¨Performance Characteristics and Limitations of the Available Assays for the Detection and Quantitation of Monoclonal Free Light Chains and New Emerging Methodologies¨, the authors realize a review of the methodologies new and essential for detecting and quantifying FLC. The relevance of FLC detection is crucial for some human pathologies. Also, the authors discuss novel methods that impact FLC detection. Nevertheless, the manuscript needs some improvements before acceptance.
1. Include a timeline (Figure 1) of the assays mentioned in the introduction. This could be nice, considering the authors use sufficient lines about it.
2. A figure explaining the FLC protein could also be relevant., including the kappa or lambda meaning.
3. Please include in Figure 2 the reference when the image was taken.
4. In Table 1, include the appropriate references.
5. A new table with the advantages and disadvantages of the enlisted methodologies is necessary, including, for example, the future directions if they are applied.
6. In Figure 2, the legend of x axis is missing.
Author Response
Dear Reviewer Two,
Thank you very much for taking the time to review this manuscript and providing constructive comments. We have addressed your suggestions and feel that these have helped us to enhance the quality of our review article.
Reviewer Two Comments:
In the manuscript entitled ¨Performance Characteristics and Limitations of the Available Assays for the Detection and Quantitation of Monoclonal Free Light Chains and New Emerging Methodologies¨, the authors realize a review of the methodologies new and essential for detecting and quantifying FLC. The relevance of FLC detection is crucial for some human pathologies. Also, the authors discuss novel methods that impact FLC detection. Nevertheless, the manuscript needs some improvements before acceptance.
- Include a timeline (Figure 1) of the assays mentioned in the introduction. This could be nice, considering the authors use sufficient lines about it.
We have included this as Figure 1 in the revised manuscript.
- A figure explaining the FLC protein could also be relevant., including the kappa or lambda meaning.
We have included a figure demonstrating why FLC specific reagents only measure FLC. This is Figure 2 in the revised manuscript.
We have also included the following paragraph about the clinical relevance of monoclonal FLC in patients with AL amyloidosis and multiple myeloma.
“In AL amyloidosis it is the aggregation and deposition of the structurally abnormal light chains that leads to progressive organ involvement. Therefore assays that can detect and monitor changes in the levels of monoclonal light chains are key in the detection and monitoring of patients with this rare plasma cell disorder. Excess production of monoclonal FLC can also lead to direct organ toxicity in patients with multiple myeloma as high levels of circulating monoclonal FLC can cause renal impairment due to cast nephropathy. The risk of myeloma-induced renal impairment increases as the level of circulating monoclonal FLC increases and therefore tests that can rapidly identify and quantitate monoclonal light chains are crucial for the early identification of acute renal impairment due to multiple myeloma, which is most commonly due to cast nephropathy, and is rarely seen with FLC levels <500mg/L (12, 13). The identification and prompt initiation of systemic anti-myeloma treatment in patients with severe renal impairment due to cast nephropathy is essential to maximise the chances of recovery renal function, which can be achieved in over half of affected patients if rapid reduction of the monoclonal light chain level is achieved (14).”
We have added an additional statement to this sentence stating the monoclonal free light chains will be of either be kappa or lambda isotype:
“In 80-85% of patients with multiple myeloma and MGUS, the monoclonal protein produced by the clonal plasma cell population produces an intact immunoglobulin monoclonal protein, most commonly IgG, and 15-20% of patients have a FLC only monoclonal protein, which may be of either free kappa or free lambda light chain isotype, without a corresponding heavy chain is produced (4-7) .”
- Please include in Figure 2 the reference when the image was taken.
This is Figure 4 in the revised manuscript and we have added the reference details to the figure legend.
- In Table 1, include the appropriate references.
We have added the appropriate references to Table 1.
- A new table with the advantages and disadvantages of the enlisted methodologies is necessary, including, for example, the future directions if they are applied.
We have added a table summarising the available methodologies and their advantages and disadvantages. This is Table 2 in the revised manuscript.
- In Figure 2, the legend of x axis is missing.
We have added the missing figure legend. This is now figure 4 in the revised manuscript.
Thank you for taking the time to review this revised manuscript and for considering it for publication in Antibodies.
Yours sincerely
Dr Hannah Giles and Professor Karunanithi
Reviewer 3 Report
Comments and Suggestions for Authors
“Performance Characteristics and Limitations of the Available Assays for the Detection and Quantitation of Monoclonal Free Light Chains and New Emerging Methodologies” by Hannah V Giles, Kamaraj Karunanithi. (Manuscript ID: antibodies-2877492)
This manuscript reviewed several commercially available testing assays and some emerging new methodologies for the detection and quantitation of free light chains in the urine or serum of patients with suspected or proven plasma cell disorders. The strengths and limitations of these assays were discussed. The topic of the manuscript is interesting and therapeutically relevant. However the manuscript needs to address a number of issues prior to its acceptance for publication:
1) First of all, the authors need to give more background explanation on the molecular mechanisms of generating free light chain in plasma cell dyscrasias. What are the severity and the relationship of detected free light chains with the disease progression/stages within a broad range of the disorders? This kind of information would help readers to understand the meanings of the assay results.
2) The authors could convert Figure 1 into a cartoon drawing figure for describing each methodologies/assays discussed in the article.
3) A summary table for pros and cons of all FLC detection methodologies is needed. The usages of each assay for various disorders should be included.
4) The light chain N-linked glycosylation (Page 7, line 300) is interesting . Does it occur in the variable region (CDRs) or constant regions? Is N-glycosylation also detected in normal sera? Besides AL amyloidosis, is light chain N-glycosylation detectable in other plasma cell disorders as well? In Figure 2 (Page 8, line 309), titles and scales for X-axis are missing.
5) The syntax and references of the manuscript needs to be checked carefully before resubmission: “unique mass due it having…” (Page 6, line 237), “protein can analyzed by …” (Page 6, line 241), reference #91 missed out journal name, page number (Page 13, line 612)…
Comments on the Quality of English LanguagePlease check syntax and references prior to resubmission.
Author Response
Dear Reviewer Three,
Thank you very much for taking the time to review this manuscript and providing constructive comments. We have addressed your suggestions and feel that these have helped us to improve the overall quality of our review article.
Reviewer three comments and responses:
“Performance Characteristics and Limitations of the Available Assays for the Detection and Quantitation of Monoclonal Free Light Chains and New Emerging Methodologies” by Hannah V Giles, Kamaraj Karunanithi. (Manuscript ID: antibodies-2877492)
This manuscript reviewed several commercially available testing assays and some emerging new methodologies for the detection and quantitation of free light chains in the urine or serum of patients with suspected or proven plasma cell disorders. The strengths and limitations of these assays were discussed. The topic of the manuscript is interesting and therapeutically relevant. However the manuscript needs to address a number of issues prior to its acceptance for publication:
- First of all, the authors need to give more background explanation on the molecular mechanisms of generating free light chain in plasma cell dyscrasias. What are the severity and the relationship of detected free light chains with the disease progression/stages within a broad range of the disorders? This kind of information would help readers to understand the meanings of the assay results.
We have provided additional information about FLC production in the first paragraph of the introduction, which now reads as follows:
“Plasma cell dyscrasias encompass a broad range disorders, from the pre-malignant condition MGUS to AL amyloidosis, which is typically associated with a low-tumour burden, and malignant multiple myeloma in which the tumour burden is much higher (1, 2). Plasma cells secrete antibodies and in most patients with a plasma cell dyscrasia, a monoclonal immunoglobulin is detectable in the serum and/or urine due to secretion of a single type of antibody by the clonal plasma cell population. A typical antibody is composed of two identical heavy chains and two identical light chains and even in health, the light chains are secreted in slight excess in comparison to heavy chains, which results in low levels of circulating polyclonal light chains. However, in patients with plasma cell disorders the FLC are frequently produced in significant excess and the resultant circulating monoclonal light chains can play an important role in the clinical sequelae of some of the plasma cell disorders, such as AL amyloidosis. Due to somatic hypermutation and VDJ rearrangement, the monoclonal immunoglobulin produced by each patients plasma cell dyscrasia has a unique structure and biochemical properties, which enable it to be used as a biomarker to assess for signs of progression in collaboration with assessments for end organ damage and also to track the response to clonally-directed treatment (3).”
We have added the following information about FLC levels in patients with acute kidney injury:
“The risk of myeloma-induced renal impairment increases as the level of circulating monoclonal FLC increases and therefore tests that can rapidly identify and quantitate monoclonal light chains are crucial for the early identification of acute renal impairment due to multiple myeloma, which is most commonly due to cast nephropathy, and is rarely seen with FLC levels <500mg/L (12, 13). The identification and prompt initiation of systemic anti-myeloma treatment in patients with severe renal impairment due to cast nephropathy is essential to maximise the chances of recovery renal function, which can be achieved in over half of affected patients if rapid reduction of the monoclonal light chain level is achieved (14).”
The SLiM CRAB criteria which include the criterion of a serum FLC ratio ≥100 were already described.
We have described the BJP criteria for differentiating between MGUS and multiple myeloma in the following paragraph:
“Although many centres have transitioned over to using serum FLC testing in place of BJP assessments in routine practice, it is important to acknowledge that the monoclonal protein size criteria for the differentiation between MGUS and multiple myeloma is based on BJP concentration being ≥500mg/24 hours (28) and that there is no recommended serum FLC level that by definition differentiates between MGUS and myeloma. In centres where BJP analysis isn't regularly done, it's crucial to establish clear guidelines for assessing the bone marrow in patients with abnormal FLC results. This is important even if there's no organ damage, to distinguish between MGUS and myeloma based on the percentage of plasma cell infiltration in the bone marrow. It is crucial to differentiate between these conditions to ensure that the follow-up intervals are appropriate given the differential risk of progression to symptomatic multiple myeloma between these two conditions (0.5-1% per year for MGUS versus 10% per year for smouldering multiple myeloma) (42).”
- The authors could convert Figure 1 into a cartoon drawing figure for describing each methodologies/assays discussed in the article.
Figure 1 in the first version of the article is now Figure 3. We have added an additional figure (Figure 2) which describes why FLC assays specifically measure FLC and not total light chains.
- A summary table for pros and cons of all FLC detection methodologies is needed. The usages of each assay for various disorders should be included.
We have added a table listing the advantages and disadvantages of the methods approved for the detection and monitoring of monoclonal light chains in patients with multiple myeloma and related plasma cell disorders. This is Table 2 in the revised article.
- The light chain N-linked glycosylation (Page 7, line 300) is interesting . Does it occur in the variable region (CDRs) or constant regions? Is N-glycosylation also detected in normal sera? Besides AL amyloidosis, is light chain N-glycosylation detectable in other plasma cell disorders as well? In Figure 2 (Page 8, line 309), titles and scales for X-axis are missing.
We have expanded the section on glycosylation to answer the questions above and this paragraph now reads as follows:
“Light chain N-linked glycosylation of the variable region of monoclonal light chains is a post-translational modification which is easily identified using the intact light chain assays based on the presence of polytypic peaks with an increased mass-to-charge ratio compared to the typical mass-to-charge range for kappa and lambda light chains (Figure 4) (97). Light chain N-linked glycosylation has been shown to be found more commonly in patients with AL amyloidosis in comparison to patients with multiple myeloma and MGUS (98, 99) and has therefore emerged as an area of interest which may assist in the earlier identification of patients with AL amyloidosis. In contrast to heavy chains, in which glycosylation is a normal feature, light chain glycosylation has not been observed in serum samples from healthy donors which have been used as control samples in the development of these assays. The presence of N-linked glycosylation was also reported to be associated with an increased risk of progression from MGUS to a symptomatic plasma cell disorder in a study by Dispenzieri et al (100), however there is only limited data on its prognostic significance in patients undergoing treatment for symptomatic plasma cell disorders (101).”
We have added the missing labels for this figure, which is Figure 4 in the revised manuscript.
5) The syntax and references of the manuscript needs to be checked carefully before resubmission: “unique mass due it having…” (Page 6, line 237), “protein can analyzed by …” (Page 6, line 241), reference #91 missed out journal name, page number (Page 13, line 612)…
We have checked and corrected the formatting of the references.
We have corrected the sentences on page six, so they now read as follows:
"The intact light chains assays utilise the principal that the light chain of the monoclonal protein will have a specific isotype and unique mass due to its unique amino acid sequence which helps to track the monoclonal protein during the course of treatment. "
“Samples undergo an immune enrichment step to remove non-immunoglobulin proteins, then captured immunoglobulins are eluted and reduced, so that the light chain portion of the monoclonal protein can be analysed by mass spectrometry (Figure 3) .”
Thank you for taking the time to review this revised manuscript and for considering it for publication in Antibodies.
Yours sincerely
Dr Hannah Giles and Professor Karunanithi
Round 2
Reviewer 2 Report
Comments and Suggestions for Authors
The authors respond to any questions or attend to reviewers' suggestions. A minor suggestion:
In Figure 2, all legends (i.e., light blue and light green) must be replaced with other colors or bold letters to be noticeable. People with vision problems could not see this legend.
Author Response
Dear Reviewer Two,
Thank you for taking the time to review our revised manuscript and we are grateful for your further suggestion to help us improve the presentation of the figures within the manuscript.
In response to your suggestion, we have changed the labels in Figure 2 to black text, increased the font size by one size and also made the labels bold text.
Thank you once again for taking the time to review this second revision of our review article entitled 'Performance Characteristics and Limitations of the Available Assays for the Detection and Quantitation of Monoclonal Free Light Chains and New Emerging Methodologies' for publication in Antibodies.
Yours Sincerely
Dr Hannah Giles and Professor Karunanithi